# TIM, a targeted insertional mutagenesis method utilizing CRISPR/Cas9 in *Chlamydomonas reinhardtii*

Tyler Picariello[1☯¤], Yuqing Hou[1☯], Tomohiro Kubo[2], Nathan A. McNeill[1], Haru-aki Yanagisawa[3], Toshiyuki Oda[2], George B. Witman[1] *

**1** Division of Cell Biology and Imaging, Department of Radiology, University of Massachusetts Medical School, Worcester, Massachusetts, United States of America, **2** Department of Anatomy and Structural Biology, Interdisciplinary Graduate School, University of Yamanashi, Chuo, Yamanashi, Japan, **3** Graduate School of Medicine, The University of Tokyo, Tokyo, Japan

☯ These authors contributed equally to this work.
¤ Current address: Sanofi-Genzyme, Framingham, Massachusetts, United States of America
* george.witman@umassmed.edu

**Data Availability Statement:** The protocol for TIM has been deposited in protocols.io and may be accessed at http://dx.doi.org/10.17504/protocols.

## Abstract

Generation and subsequent analysis of mutants is critical to understanding the functions of genes and proteins. Here we describe TIM, an efficient, cost-effective, CRISPR-based targeted insertional mutagenesis method for the model organism *Chlamydomonas reinhardtii*. TIM utilizes delivery into the cell of a Cas9-guide RNA (gRNA) ribonucleoprotein (RNP) together with exogenous double-stranded (donor) DNA. The donor DNA contains gene-specific homology arms and an integral antibiotic-resistance gene that inserts at the double-stranded break generated by Cas9. After optimizing multiple parameters of this method, we were able to generate mutants for six out of six different genes in two different cell-walled strains with mutation efficiencies ranging from 40% to 95%. Furthermore, these high efficiencies allowed simultaneous targeting of two separate genes in a single experiment. TIM is flexible with regard to many parameters and can be carried out using either electroporation or the glass-bead method for delivery of the RNP and donor DNA. TIM achieves a far higher mutation rate than any previously reported for CRISPR-based methods in *C. reinhardtii* and promises to be effective for many, if not all, non-essential nuclear genes.

## Introduction

The green alga *Chlamydomonas reinhardtii* provides an excellent model system for the study of cilia/flagella, cell cycle, photosynthesis, and mitochondria [1, 2]. A commonly used method for the generation of mutants in this organism is insertional mutagenesis [3–6]. This method relies on the transformation of *C. reinhardtii* strains with exogenous double-stranded DNA (commonly an antibiotic-resistance gene) that integrates randomly into the genome, causing a mutation at the insertion site. The integrated exogenous DNA confers antibiotic resistance, which facilitates screening of transformants and identification of the gene containing the

io.bdcki2uw. All relevant data are within the paper
and its Supporting Information files.

**Funding:** This work was supported by National
Institutes of Health (https://www.nih.gov/) grants
R37 GM030626 and R35 GM122574 (to G.B.W.);
by the Robert W. Booth Endowment at the
University of Massachusetts Medical School
(https://www.umassmed.edu/) (to G.B.W.); by the
Japan Society for the Promotion of Science
(https://www.jsps.go.jp/english/) KAKENHI grants
17H05057 (to T.O.) and 17K15115 and 19K16123
(to T.K.); and by the Takeda Science Foundation
(https://www.takeda-sci.or.jp/index.html) (to T.O.
and T.K.). The funders had no role in study design,
data collection and analysis, decision to publish, or
preparation of the manuscript.

**Competing interests:** The authors have declared
that no competing interests exist.

insertion [7–10]. Rescue of insertional mutants with genetically engineered constructs then allows interrogation of specific residues, domains, etc.

Recently, a genome-wide mutant library for *C. reinhardtii* was developed using insertional mutagenesis [8, 11]. While this has broadly benefited genetic analysis in this organism, several drawbacks exist. First, the project has not yet achieved universal coverage of the genome. Second, some genes only have insertions in non-coding regions which may not result in a null mutation, complicating downstream analyses. Third, the mutants were created using a strain that has impaired motility and lacks a cell wall, which reduces plating efficiencies and precludes isolation of highly pure flagella. Finally, for some experiments there may be a need to have mutations in a different genetic background, or double mutants, either of which would require extensive genetic crossing to accomplish.

The clustered regularly interspaced short palindromic repeats (CRISPR)/CRISPR-associated protein 9 (Cas9) system has greatly facilitated our ability to edit genomes in many organisms [12, 13]. The first report using CRISPR/Cas9 in *C. reinhardtii* demonstrated that Cas9 and a single guide RNA expressed from a plasmid electroporated into a strain lacking a cell wall was active, but only a single stable transformant with a deletion consistent with Cas9 editing was obtained from an initial pool of $1.6 \times 10^9$ cells [14]. This raised concerns that Cas9 was toxic in *C. reinhardtii*. More recent studies, exploring different approaches, have clearly demonstrated that CRISPR/Cas9 is useful for targeted disruption of genes in *C. reinhardtii* [15–22]. However, some of the published methods have low efficiencies of achieving the desired edit, making it laborious to identify cells in which a gene has been successfully targeted, especially in the absence of a selectable marker. Additionally, some methods can be applied only to certain genes or strains.

To establish a universal, highly efficient, and relatively simple CRISPR/Cas9-based gene-editing protocol for targeted disruption of genes, we started with the method of Shamoto et al. [23], who briefly described a protocol in which about 55% of picked colonies had insertional mutations in the targeted gene. This is an efficiency that, as far as we know, is higher than any previously reported. To understand which aspects of the protocol are critical for achieving such efficiency, and whether the protocol could be further improved, we first reproduced their results. We then carried out a series of pair-wise experiments; in each experiment, a single parameter was varied, the results assessed, and any perceived improvement incorporated into the working protocol. Our aim was not to achieve statistical significance for any one variable, but to define the important conditions enabling a simple, reliable, flexible, and robust method for targeted insertional mutagenesis, which we term TIM. As we demonstrate below, TIM is applicable to cell-walled *C. reinhardtii* strains, can be applied to cells grown on agar plates or in liquid medium, can utilize different antibiotic-resistant selectable markers, works with electroporation or the glass-bead method for delivery of macromolecules into the cells, yields mutation efficiencies as high as 90%, and even can be used to generate double mutants. We have used the TIM method to target six different genes, and in each case we achieved a high rate of mutagenesis. Thus, the method promises to be effective for many, if not all, non-essential nuclear genes.

## Results

In the TIM method (Fig 1), *C. reinhardtii* cells are treated with autolysin to remove their cell walls. A Cas9-guide RNA (gRNA) ribonucleoprotein (RNP) together with exogenous double-stranded (donor) DNA containing gene-specific homology arms and an integral antibiotic-resistance gene are then delivered to the cells using electroporation or the glass-bead method. The RNP generates a double-strand break (DSB) at the target site, which facilitates insertion of

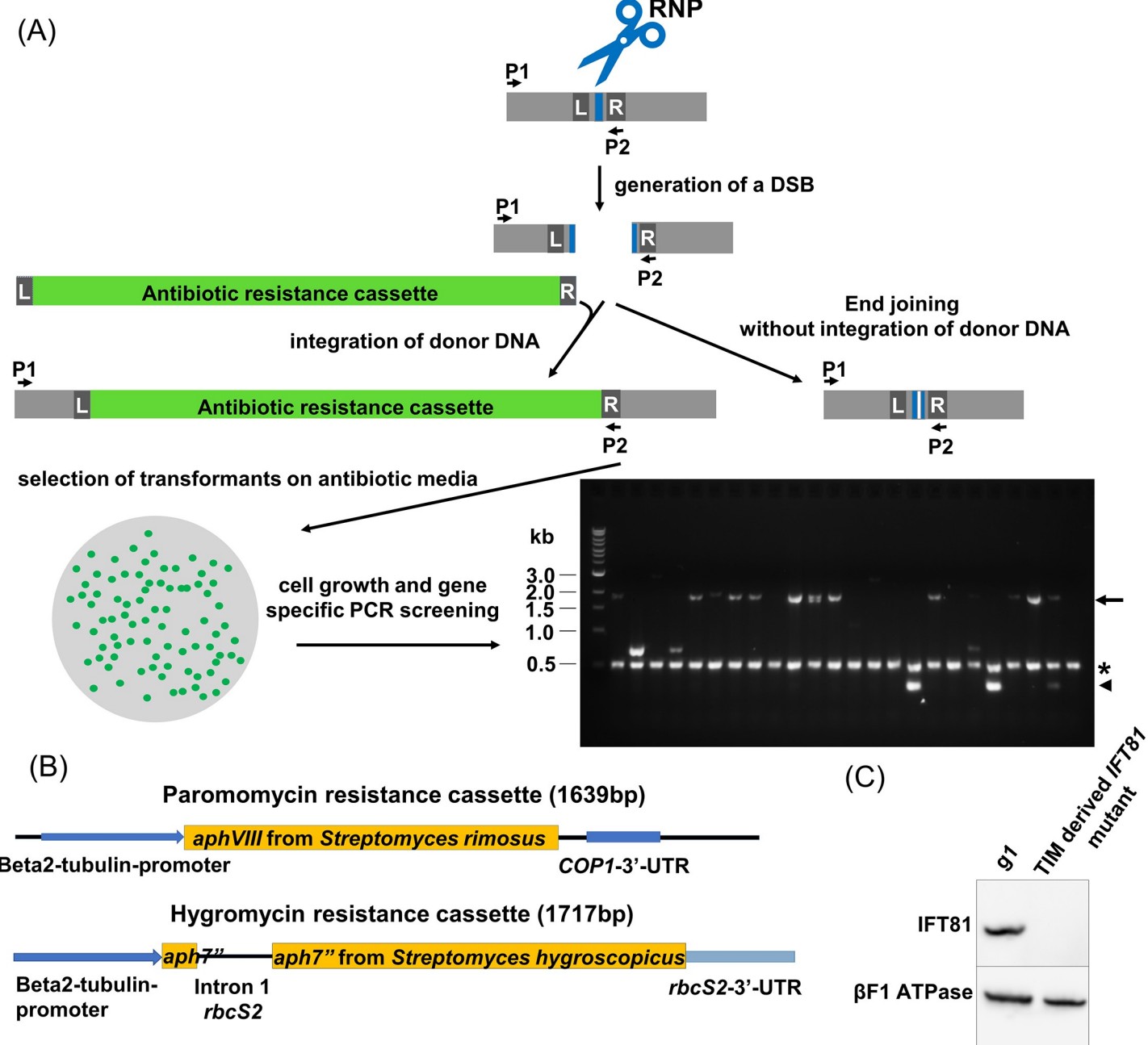

**Fig 1. An overview of the TIM (targeted insertional mutagenesis) method.** (A) Schematic overview of TIM. Following delivery of Cas9/gRNA RNP and double-stranded donor DNA (antibiotic-resistance cassette) into cells by electroporation or the glass-bead method, the RNP causes a double-stranded break (DSB) at the target site (blue box). DSBs are repaired with or without integration of the donor DNA. The left pathway results in insertional mutations, whereas the right pathway may generate insertion/deletion (indel) mutations (small white box within blue box, third row). As with other insertional mutagenesis approaches, donor DNA also can insert randomly in the genome, resulting in antibiotic-resistant cells that may not have a mutation at the target site. Cells are then selected for presence of the donor DNA via growth on antibiotic-specific media. Finally, antibiotic-resistant cells are screened via PCR to identify cell lines with mutations at the target site. L and R, left and right homology arms of the donor DNA, or genomic regions corresponding to left and right homology arms of the donor DNA. Short black arrows (P1, P2) indicate the primers used for screening. In the typical agarose gel shown (in this case, the targeted gene was *FAP70*), the asterisk and arrowhead on the right side of the gel indicate positive control (in this case *FUS1*) and wild-type target-specific PCR bands, respectively. Absence of a wild-type target-specific band indicates a mutation at the target site; in most cases here this is accompanied either by the appearance of an ~2-Kb band (arrow on the right side of the gel), suggesting insertion of a single copy of the antibiotic-resistance gene (~1.7 Kb), or by lack of any detectible gene-specific band, suggesting a large insertion/deletion at the target site. Presence of a wild-type target-specific band indicates that the cell line was not mutated at the target site; the antibiotic-resistance gene has inserted into the genome somewhere other than at the target site. Mutants generated by the right-hand pathway (no insertion of donor DNA) would not be selected for unless the donor DNA inserted into another site; in the latter case, mutants with small indels would appear to be wild type in the PCR screen because the PCR product would be at or near wild-type length. Relative to the

target site, the exact positions of the primers and of genomic regions corresponding to the homology arms will vary when targeting different genes. (B) Schematic of the antibiotic-resistance cassettes used. Gene-specific homology arms of varying lengths were added to the 5'- and 3'-ends of these cassettes. (C) Representative western blot of whole-cell extracts of g1 (wild type) and an *ift81* mutant generated by TIM. Absence of an IFT81 band in the mutant strain confirms disruption of the *IFT81* gene. ATP synthase β subunit (βF1 ATPase) was probed as a loading control.

the exogenous DNA at that site (although insertions can occur randomly in other regions of the genome as well). If the antibiotic-resistance gene remains intact, the insertion confers antibiotic resistance to the cells. The insertion may be accompanied by concatenation of the cassette, deletion of genomic DNA, or reorganizations of the target site, all of which are common to insertional mutagenesis techniques [6]. Alterations resulting from insertion at the target site are then easily identified by PCR, with the mutation rate calculated as the number of colonies carrying an insertion at the target site divided by the total number of drug-resistant colonies picked for analysis.

Shamoto et al. [23] did not specify the conditions used to grow the cells in their experiment, but correspondence with them established that growth was on agar plates in continuous light. Initially, we duplicated their method by targeting exon 2 of *FAP70* in strain CC-125 (137c), which they used, and in strain CC-5415 (g1) [24]; both of these strains have cell walls. In four different experiments, we achieved mutation efficiencies of 42% and 40% for 137c and 50% and 62% for g1 (Table 1, rows 1–4). These results are consistent with the 55% mutation rate reported by Shamoto and colleagues. Therefore, it is likely that TIM also will work well with other strains. Our subsequent experiments were carried out using the g1 strain.

Cultures grown in continuous light are likely to contain cells in various phases of the cell cycle, only one of which may be favorable to achieving high mutation rates. To determine if this is a critical parameter, we grew g1 cells on a 14 h light:10 h dark cycle and targeted the *FAP70* gene during the middle of the light cycle. In two separate experiments, we obtained mutation efficiencies of 96% and 85% (Table 1, row 5). Therefore, cells grown synchronously and used in the middle of the light cycle yield efficiencies at least as high as those grown on continuous light. Subsequent experiments were carried out using cells grown on this light: dark cycle.

To determine if growth on agar plates was a stringent parameter for efficient mutagenesis, we grew cells in liquid TAP medium with shaking for two days. In each of two experiments, we inoculated flasks with three different concentrations of cells so that the cultures would have different cell densities (ranging from $6.6 \times 10^5$ to $1.3 \times 10^7$ cells/mL) when harvested for mutagenesis. The resulting mutagenesis rates varied from 36% to 70%, with slightly higher mutagenesis rates for cells harvested at densities of about $3.8 \times 10^6$ cells/mL or higher (Table 1, rows 6–11). Therefore, TIM is efficient with cells grown on agar plates or in liquid culture, and culture cell density does not appear to be critical within the range tested. However, harvesting cells grown on agar plates was easier, so we used this manner of growth in all other experiments reported here.

Removal of the cell wall prior to transformation of cell-walled cells with plasmids by electroporation is not necessary [e.g., 25–27]. To determine if cell-wall removal is required for delivery of RNPs into cells by electroporation during TIM, we targeted *FAP70* in g1 cells treated with and without autolysin. In the absence of autolysin treatment, we achieved only a 3% mutation rate, versus an 87% mutation rate for the autolysin-treated cells (Table 1, rows 12–13). Therefore, cell-wall removal is critical for efficient mutagenesis using the TIM method. In all experiments presented here, except that of row 12, the efficiency of cell-wall removal was ~50% or greater. In addition, in all paired experiments, the same batch of autolysin-treated cells was utilized for both experiments.

**Table 1. Summary of experiments and results*.**

| Row | Parameters | Conditions† | Mutation efficiency (# mutant colonies/# of colonies screened) | | |
|---|---|---|---|---|---|
| | | | repeat 1 | repeat 2 | repeat 3 |
| 1 | Cell line‡ | 137c | 42% (40/96) | | |
| 2 | | 137c | 40% (37/94) | | |
| 3 | | g1 | 50% (45/90) | | |
| 4 | | g1 | 62% (60/96) | | |
| 5 | Light:dark cycle | 14 h light:10 h dark cycle | 96% (46/48) | 85% (39/46) | |
| 6 | Liquid culture cell density | 6.6E5 | 48% (22/46) | | |
| 7 | | 3.2E6 | 38% (18/48) | | |
| 8 | | 3.84E6 | 70% (32/46) | | |
| 9 | | 2.36E6 | 36% (16/45) | | |
| 10 | | 4.56E6 | 67% (31/46) | | |
| 11 | | 1.33E7 | 64% (30/47) | | |
| 12 | Autolysin treatment | No | 3% (1/40) | | |
| 13 | | Yes | 87% (41/47) | | |
| 14 | Homology arm length | 50 bp | 85% (39/46) | 67% (32/48) | |
| 15 | | 25 bp | 76% (34/45) | 63% (30/48) | |
| 16 | | 0 bp | | 42% (20/48) | |
| 17 | Selectable marker cassette | Paromomycin | 78% (36/46) | | |
| 18 | | Hygromycin | 84% (38/45) | | |
| 19 | gRNA quality | Freshly made | 87% (41/47) | | |
| 20 | | Frozen and thawed | 49% (23/47) | | |
| 21 | Cell density during electroporation | 3.50E7 cells/ul | 68% (32/47) | | |
| 22 | | 1.76E8 cells/ul | 87% (41/47) | | |
| 23 | Delivery of RNP and donor DNA | Electroporation | 87% (41/47) | | |
| 24 | | Glass beads | 47% (21/45) | | |
| 25 | Gene plus gRNA and cassette design | IFT81 gRNA 1 | 42% (19/45) | 36% (16/45) | |
| 26 | | FAP70 | 85% (39/46) | 67% (32/48) | |
| 27 | | IFT81 gRNA 2 | 40% (19/48) | | |
| 28 | | FAP70 | 63% (30/48) | | |
| 29 | | IFT81 gRNA 1 | 30% (14/47) | | |
| 30 | | IFT81 gRNA 2 | 52% (24/46) | | |
| 31 | | MOT17 | 93% (41/44) | | |
| 32 | | CDPK13 | 47% (20/43) | | |
| 33 | | FAP70 | 78% (36/46) | | |
| 34 | | CEP131 gRNA-1 no homology arms | 41% (19/46) | | |
| 35 | | CEP131 gRNA-2 no homology arms | 30% (14/46) | | |
| 36 | | IFT43 no homology arms | 89% (40/45) | | |
| 37 | Double mutations | IFT81 PARO/ FAP70 HYG | 15% (6/40) | 3% (1/38) | 6% (3/48) |

*Experiments in the same color are comparable with only one parameter changed between them.

†Except as noted, all experiments used autolysin treatment followed by electroporation to deliver the RNP and donor DNA with homology arms into the cells.

‡For these experiments only, cells were grown in continuous light.

Shamoto et al. [23] used donor DNA containing gene-specific homology arms of 50 bp, reasoning that this would increase the likelihood of insertion of the antibiotic-resistance cassette at the target site. To determine how the length of the homology arms affects mutation efficiency, we targeted *FAP70* using donor DNA with 50-bp and 25-bp homology arms. In two

separate tests, we achieved 85% and 67% mutation rates for donor DNA with 50-bp homology arms vs. 76% and 63% for donor DNA with 25-bp homology arms (Table 1, rows 14 and 15). In the second of the above experiments, we further compared donor DNA completely lacking homology arms, which yielded a 42% mutation rate (Table 1, row 16). Therefore, the presence of homology arms appears to be beneficial, but the homology arms are not absolutely required for efficient mutagenesis.

Shamoto et al. [23] used an antibiotic-resistance cassette containing the *Streptomyces rimosus aphVIII* gene and selected for insertional mutants using paromomycin. To determine if choice of antibiotic-resistance gene affects mutation efficiency, we compared donor DNAs containing the *aphVIII* gene vs. the *aph7"* gene from *S. hygroscopicus* conferring hygromycin resistance (Fig 1B). We achieved a 78% mutation rate using the former, compared to 84% with the latter (Table 1, rows 17 and 18). These data indicate that the high mutation efficiency of TIM is not dependent on use of a specific antibiotic-resistance gene and suggests that investigators may be able to use a variety of selectable markers with a high degree of success.

To determine how handling and storage of the gRNA affected mutation rate, we carried out an experiment in which we targeted *FAP70* with fresh gRNA vs. gRNA that had undergone several freeze/thaw cycles. We obtained mutation rates of 87% and 49% respectively (Table 1, rows 19 and 20), underscoring the importance of using fresh gRNA to achieve maximum efficiency.

Shamoto et al. [23] provided no recommendation on cell density during electroporation. To determine if this was an important variable, we targeted *FAP70* by electroporation at cell densities in the cuvette of $3.5 \times 10^7$ and $1.8 \times 10^8$ cells/ml and obtained mutagenesis rates of 68% and 87%, respectively (Table 1, rows 21 and 22). Therefore, efficient mutagenesis can be obtained at cell densities over at least the five-fold range tested. In subsequent experiments, we used cell densities that we estimated to be near the upper limit of that range.

All of the above experiments were carried out by delivering the gRNA and donor DNA to the cells by electroporation. However, not all labs have access to an electroporator, and it may not be possible to exactly match electroporation conditions with electroporators from different manufacturers. An alternative, and widely used, method for delivery of macromolecules into *C. reinhardtii* cells is the glass-bead method [28]. To determine how the efficiency of TIM is affected by the method of delivery, we carried out an experiment comparing electroporation to a slightly modified version of the glass-bead method. Targeting *FAP70*, we achieved an 87% mutation rate with the former and a 47% mutation rate with the latter (Table 1, rows 23 and 24). Therefore, although the efficiency with the glass-bead method may be lower than with electroporation, the glass-bead method provides a second, highly economical, method for TIM when electroporation is not readily available.

To clarify the types of mutations created by TIM, we randomly selected seven *fap70* mutant strains that appeared to have ~2-kb bands when the target site was screened by PCR (Fig 1A). These mutant strains were generated using donor DNA containing the paromomycin-resistance cassette and 50-bp homology arms. In wild type, sequence corresponding to the left homology arm was 27 bps upstream of the protospacer adjacent motif (PAM); sequence corresponding to the right homology arm was 3 bps downstream from the PAM. DNA from the PCR bands was purified, sequenced (S1 Appendix), and aligned with donor DNA and wild-type sequences (S2 Appendix). The sequences from five of the strains (C1, C2, C3, D1, and D4) were the same as that of wild type except that the 33 bps between the sequences corresponding to the homology arms were replaced with sequence corresponding to an intact paromomycin-resistance cassette, as expected from homology-directed repair (HDR). The 5' end of the sequence from strain C6 matched that of wild type until 2 bps upstream of the PAM, at which point a partial cassette, immediately preceded by a C, was inserted. Compared to an

intact cassette, this partial cassette lacked the first 167 bps at the 5' end of the cassette, as well as the last 165 bps at the 3' end of the cassette, downstream of the *COP1*-3'-UTR (Fig 1B). Following the sequence corresponding to the partial cassette, there was a 16-bp insertion. The remainder of the sequence matched the wild-type genome starting 3 bps upstream of the PAM and continuing through sequence corresponding to the right homology arm. The 5' end of the sequence from the seventh strain (D6) matched the 5' end of the wild-type sequence until 7 bps upstream of the PAM. At this point, there was a 158-bp insertion corresponding to bps 1496–1653 of the donor DNA. This was followed by sequence corresponding to a partial cassette in which the first 163 bps were deleted. The remaining portion of the cassette was followed by sequence corresponding to the right homology arm. The 167-bp and 163-bp deletions in strains C6 and D6, respectively, removed the first 121 bps (strain C6) and 117 bps (strain D6) of the beta-tubulin promoter upstream of the paromomycin-resistance gene (*aphVIII*) coding sequence. Since these strains were paromomycin resistant, it seems likely that either a) the truncated beta-tubulin promoter retained activity; b) another promoter, such as that of *FAP70*, served as the promoter for the *aphVIII* gene; or c) a functional cassette was inserted at another site in the genome. In any case, in all seven strains, donor DNA was inserted at the intended target site, resulting in an insertional mutation likely to cause disruption of the *fap70* gene.

To determine the effectiveness of TIM for genes other than *FAP70*, we tested the method on the IFT complex-B gene *IFT81*, using *FAP70* as positive control. *IFT81*, which we targeted at exon 3, was chosen for this test because the *ift81*-null mutant is well characterized [27] and an excellent antibody to IFT81 is available [29]. Initially, we used the same autolysin-treated preparations of cells and targeted either *IFT81* or *FAP70* using a paromomycin-resistance cassette with 50-bp gene-specific homology arms. In two experiments, we obtained efficiencies of 42% and 36% for *IFT81* vs. 85% and 67% for *FAP70* (Table 1, rows 25 and 26). The design of the gRNA is undoubtedly important for efficient mutagenesis, so we next tried a different gRNA for *IFT81*; in this case we obtained mutation efficiencies of 40% for *IFT81* vs. 63% for *FAP70* (Table 1, rows 27 and 28). Finally, we compared the two *IFT81* gRNAs directly, and obtained efficiencies of 30% for the first and 52% for the second (Table 1, rows 29 and 30). It seems likely that the lower mutation rates obtained for *IFT81* represent lower efficiencies of the *IFT81* gRNAs, although one cannot rule out that some genes are inherently less efficiently targeted than others by TIM. In any case, the efficiencies obtained for *IFT81* were sufficiently high to easily identify insertional mutants. The *ift81* mutants that we generated exhibited a palmelloid phenotype–i.e., they failed to hatch from the mother cell wall–matching the phenotype previously reported for an *ift81* null mutant [27]. No bands corresponding to IFT81 were detected in western blots of these mutants, confirming disruption of the *IFT81* gene by TIM (Fig 1C).

To further assess the effectiveness and broader applicability of TIM, we targeted four other genes: *MOT17*, *CDPK13*, and *CEP131*, for which no mutants have yet been reported, and *IFT43*. *MOT17* and *CDPK13* were targeted in the same autolysin-treated batch of cells using *FAP70* as positive control and donor DNA having 50-bp homology arms; mutagenesis efficiencies of 93% and 47% were obtained for *MOT17* and *CDPK13*, respectively, vs. 78% for *FAP70* (Table 1, rows 31–33). *CEP131* was targeted using two different guide RNAs and donor DNA without homology arms, which yielded mutation rates of 41% and 30% (Table 1, rows 34 and 35). *IFT43* also was targeted using donor DNA without homology arms, resulting in a mutation rate of 89% (Table 1, row 36). Therefore, TIM appears to be effective with a wide range of genes. Further characterization of these mutants is ongoing.

The high mutation efficiencies that we obtained with different genes suggested that it might be possible to target two different genes in a single experiment. To test this, we used RNPs and

donor DNAs for *IFT81* (Chromosome 7) and *FAP70* (Chromosome 6). These experiments used paromomycin- and hygromycin-resistance cassettes with 50-bp homology arms for *IFT81* and *FAP70*, respectively, and antibiotic-resistant cells were selected on agar plates containing both paromomycin and hygromycin. In three independent experiments, we were able to generate dual mutants with efficiencies of 15%, 3%, and 6% (Table 1, row 37). As expected, the mutation rates for generation of double mutants was much lower than that achieved when targeting only a single gene, but the results clearly demonstrate that two independent genes can be targeted simultaneously using TIM.

## Discussion

Efficient mutagenesis of specific nuclear genes has been a long-standing challenge in the *C. reinhardtii* research field. Here we addressed this challenge by starting with the briefly described method of Shamoto et al. [23] and varying several parameters to determine which are critical and which can be changed as might be desirable for specific circumstances. We also tested the broader applicability of TIM by targeting several different genes.

Many studies are best carried out with cell-walled strains, so we felt that it was imperative that a general method for targeted insertional mutagenesis be applicable to, and efficient for, such strains. This is indeed the case for TIM, which works well with the cell-walled strains CC-125 and CC-5415. However, we found that it was critical to remove the cell walls by autolysin treatment prior to delivery of the RNP by electroporation. Cell-wall removal by autolysin is usually, if not always, done for delivery of DNA into cells by the glass-bead method [28], but in previous studies we have found that it is not necessary for electroporetic transformation with DNA [e.g., 25–27]. It is possible that cell-wall removal is specifically required for electroporation of RNP, as opposed to DNA, into cells. In any case, we recommend that cells be used for TIM only if the autolysin treatment is effective as assessed by treatment of cells with detergent (see Materials and Methods). If the majority of cells do not lyse, then the experiment should be jettisoned and a new batch of autolysin prepared.

It also was imperative that the method be applicable to different strains, so that one could use any strain with a genetic background advantageous for a particular approach or experiment. TIM was highly efficient with the two strains tested here, so it is likely also to work well with many other strains.

The ability to choose between selectable markers is important if mutants are to be generated in strains already carrying an antibiotic-resistance gene (e.g., a previously generated mutant). We tested two different antibiotic-resistance genes–*aphVIII* from *S. rimosus* and *aph7"* from *S. hygroscopicus*–and observed similarly high mutagenesis rates with both. It is likely that other selectable markers also will work with TIM. The ability to use different antibiotic-resistance genes also allowed us to simultaneously target two different genes and then select for double mutants resistant to both antibiotics.

Surprisingly, we found that high efficiencies of insertional mutagenesis could be achieved using insertion cassettes without homology arms. Apparently, after the double-stranded break is generated by the RNP, the cell's DNA-repair machinery joins the broken ends of the chromosome to the ends of the cassette (*sans* homology arms) often enough that correctly targeted insertional mutants can be readily selected under our conditions. However, in careful comparisons, we found that the percent of antibiotic-resistant strains with correctly targeted insertions increased with the use of 25-bp homology arms and was even greater with the use of 50-bp homology arms. Therefore, we recommend including homology arms, especially when targeting a new gene for which design of the gRNA may not be optimized.

The TIM method described here accommodates considerable flexibility with regard to method of cell growth (continuous light vs. a light-dark cycle; liquid medium vs. agar plates), culture cell density, density of cells in the electroporation chamber, and method of delivering macromolecules to the cells (electroporation vs. glass-bead method). Assuming that the high efficiencies observed with the six genes targeted here extend to other genes, there may be little need for further improvement. However, because the efficiency of a new gRNA cannot be predicted with certainty, an even more effective protocol might make the difference between success and failure. Any improvement to the protocol might also result in higher efficiencies when generating double mutants. To this end, future work might examine the effect of increasing the concentration of the RNP and donor DNA, increasing the length of the homology arms on the antibiotic-resistance cassettes beyond 50 bps, or varying the ratio of RNP to donor DNA.

It also would be interesting to determine if a Cpf1 RNP [17] could replace the Cas9 RNP used in our method. Cpf1 recognizes a different PAM sequence than Cas9. If successful, the use of Cpf1 would provide researchers with a wider variety of target sites for any given gene [30]. Additionally, Cpf1 generates a staggered DSB, whereas Cas9 generates a blunt DSB. Staggered DSBs may be more efficient than blunt DSBs in exogenous DNA insertion. This could further increase the efficiency of both the single- and dual-target experiments.

In many organisms, it is possible to precisely edit (as opposed to disrupt) genes via CRISPR/Cas9-mediated HDR, and there currently is much interest in developing efficient methods for such precise gene editing in *C. reinhardtii* [e.g., 17, 18, 20]. However, if one has a *C. reinhardtii* null mutant, such as those generated by TIM, that mutant usually can be "rescued" by transformation with constructs designed to express the affected protein with engineered changes ranging from a single amino-acid substitution to addition of an affinity or fluorescent tag [e.g., 27, 31–40]. For non-essential genes, the ease with which this can be accomplished in *C. reinhardtii* makes this approach a reasonable alternative to precise gene editing by CRISPR/Cas9-mediated HDR.

TIM can result in a range of mutations in the target gene, as evidenced by the different band patterns observed in the PCR screen shown in Fig 1A. Sequencing the mutations from several *fap70* mutants that had yielded large bands in our PCR screen revealed that most had clean insertions of the donor DNA at the target site, and all had large (~1.7-kb) insertions likely to cause disruption of the gene. The *ift81* mutants that we characterized also were functionally null. However, in detailed analysis of several insertional mutants obtained from the CLiP library (8), we found that some still expressed sequence from either the N- or C-terminus of the mutated protein (40). Therefore, for any mutant created by insertional mutagenesis, it is advisable to characterize the mutant at the protein level before concluding that it is truly null.

In summary, we have tested several parameters and conditions for TIM, identifying those that are and are not critical for success. We have used TIM to target six different genes with eight different gRNAs and achieved highly efficient mutagenesis (30–90%) in each case. Therefore, similar results are likely to be obtained with many other genes. The protocol that we recommend based on these experiments is detailed in the Materials and Methods section.

## Materials and methods

### Strains and media

*C. reinhardtii* strains CC-620 (*nit1*, *nit2*, *mt*+), CC-621 (*nit1*, *nit2*, *mt*-), 137c (CC-125; *nit1*, *nit2*, *mt*+) and g1 (CC-5415; *nit1*, *agg1*, *mt*+) (Chlamydomonas Resource Center, https://www.chlamycollection.org/) were utilized in this study. Media used were: Tris-acetate-phosphate (TAP) medium [41]; M (minimal) medium I [42] modified to contain 0.0022 M $KH_2PO_4$ and 0.00171 M $K_2HPO_4$ [43]; M-N medium (modified M medium lacking $NH_4NO_3$).

## gRNA and donor DNA design

gRNAs for *IFT81* (first gRNA), *IFT43*, *MOT17*, and *CDPK13* were designed using Integrated DNA Technology's (IDT) online custom Alt-R® CRISPR-Cas9 guide RNA tool (https://www.idtdna.com/site/order/designtool/index/CRISPR_CUSTOM). gRNAs for *FAP70*, *IFT81* (second gRNA), and *CEP131* were designed using the CRISPR-direct website (http://crispr.dbcls.jp) [44]. The paromomycin-resistance cassette was amplified from the pKS-aphVIII-lox aph-VIII plasmid (Chlamydomonas Resource Center), while the hygromycin-resistance cassette was amplified from the pHyg3 plasmid [45]. All CRISPR RNAs (crRNAs) and primers used in this study are listed in S1 Table.

## Preparation of Cas9/gRNA RNP

IDT Alt-R® CRISPR-Cas9 crRNA (10 μL, 40 μM) was annealed with IDT Alt-R® CRISPR-Cas9 tracrRNA (10 μL, 40 μM) by heating to 95°C for 2 minutes and then cooling slowly to room temperature. Two microliters of annealed gRNA were incubated with 5 μg of IDT Alt-R® S.p.Cas9 Nuclease V3 in IDT RNA duplex buffer at 37°C for 15 minutes in a final volume of 10 μL.

## Autolysin preparation

Aliquots (250 μL/dish) of CC-620 and CC-621 cell cultures were separately plated on 10-cm culture dishes containing TAP medium + 1.5% agar. Two plates for each cell line were grown under a 14:10 light:dark cycle until the cells reached confluency (usually 4–7 days depending on the amount of starting cells). Cells from each strain were scraped from the agar plates, transferred to 20 mL of M-N medium, and resuspended by vortexing. Cells were collected by centrifugation at 1819 x g for 5 min, resuspended in 20 mL fresh M-N medium, and incubated in a 250-mL Erlenmeyer flask under constant light with very gentle shaking until gametes formed (usually 3–5 hours). Gametes of opposite mating types were mixed and left under constant light for 5 minutes without agitation. The mixture was centrifuged at 11,950 x g for 10 minutes at 4°C to remove cells, and the supernatant harvested, filtered through a 0.45-μm filter (Corning™ Centrifuge Tube Top Vacuum Filter 430314), and stored long-term at -80°C.

## Delivery of Cas9/gRNA RNP and donor DNA

Cells were grown on TAP plates for 3–5 days (either under 14:10 light:dark cycle or under constant light). Cells were transferred from plates to 6 mL of autolysin-containing solution (see the section above) for 1 hour at room temperature to remove their cell walls. To test the effectiveness of the autolysin treatment, a small aliquot of cells (~10 μL) was mixed with an equal volume of 0.5% Triton-X100 and observed under a phase microscope using a 16x-objective lens. If the majority of cells had undergone lysis, the autolysin treatment was considered effective. Multiple treatments with autolysin might be needed to achieve a high degree of cell-wall removal. Following cell-wall removal, cells (still in autolysin solution) were incubated at 40°C for 30 minutes with gentle agitation, a treatment that Greiner et al. (18) concluded was beneficial for processes related to CRISPR/Cas9 activity, DNA repair, DNA integration, and/or homologous recombination. To completely remove the autolysin, cells were collected by centrifugation at 1819 x g, washed once with 10 mL TAP + 2% sucrose, and then resuspended in TAP + 2% sucrose at a concentration of 2.0–7.0x10^8 cells/mL. Ten μL of Cas9/gRNA RNP and 2 μg of donor DNA were mixed with approximately 110 μL of concentrated cells to give a final volume of 125 μL. The RNP and donor DNA was then delivered into the cells either by electroporation or the glass-bead method. For electroporation, the mixture was electroporated in a

Bio-Rad 0.2-cm gap cuvette using a BTX ECM 600 at 350V, 25Ω, and 600 μF [46]. The cuvette was then incubated immediately at 16˚C for 1 hour.

For the experiment to test the effect of growing cells in liquid culture, 250-mL Erlenmeyer flasks containing about 125 mL of liquid TAP medium were inoculated with 1 mL containing 1X, 5X, or 25X number of cells. Cultures were grown for two days with a 14-hour light:10-hour dark cycle and shaking at ~100 rpm until reaching densities of $6.6 \times 10^5$ to $1.33 \times 10^7$ cells/mL. Cells were collected by centrifugation and treated with autolysin as in the previous paragraph.

For the glass-bead method [28], autolysin-treated cells, RNP, and donor DNA were transferred to a 15-mL tube with 0.3 g of glass beads (0.45–0.52-mm diameter) and vortexed immediately at top speed on a Vortex Genie 2 (Scientific Industries) for 15 seconds, rested for 10 seconds, then vortexed again at top speed for 10 seconds.

Following either of the delivery methods, the cells were transferred to 10 mL fresh TAP + 2% sucrose in a 15-mL tube and gently rocked under dim light for 24 hours at room temperature. The cells were then collected as above, mixed with 3.5 mL TAP + 0.5% agar ($< 42$˚C), and pipetted onto TAP + 1.5% agar plates containing either 10 μg/mL paromomycin, 10 μg/mL hygromycin, or both. The 0.5% agar was allowed to solidify at room temperature and the plates placed under a 14:10 light:dark cycle at 23˚C for one week.

### Genotyping and sequencing of potential mutants

Antibiotic-resistant cells were picked into 150 μL M media in 96-well plates and were grown in a 14:10 light:dark cycle for 2–4 days. Ten μL of cell culture were mixed with 50 μL of 10 mM EDTA and then heated to 100˚C for 20 minutes to generate crude genomic DNA for screening cell lines by PCR. Primer sequences used to screen the cell lines are listed in S1 Table. The housekeeping gene *RACK1* [47] or mating-type-plus specific gene *FUS1* [48] was used as positive control. Mutation rate was the percent of antibiotic-resistant cell lines in which the targeted gene was disrupted as determined by PCR (Fig 1A). NCBI and Phytozome 12 identifications for all targeted genes are listed in S2 Table.

For sequencing of *fap70* mutations, PCR bands were gel purified using the Zymoclean gel DNA recovery kit (Zymo Research) according to the manufacturer's instructions. The purified DNA was sent for sequencing using the primers FAP70 F, FAP70 R, and AphVIII-1 (S1 Table).

### Western blotting

Wild-type cells and *IFT81* mutants were grown in liquid M medium in a 24-well plate for 5 days and were harvested for preparation of whole-cell extracts. Briefly, cells were collected by spinning in an Eppendorf tabletop centrifuge at 845 x g for 5 minutes. The cell pellet was resuspended in 5x-denaturing sample buffer (50 mM Tris, pH 8.0, 160 mM DTT, 5 mM EDTA, 50% sucrose, and 5% SDS), heated to 90˚C for 5 min, then passed through a 26-gauge needle six times. Equal amounts of protein from control and test samples were subjected to SDS-PAGE and transferred to Immobilon P (EMD Millipore, Billerica, MA) membrane for subsequent analysis. Membranes were probed with the following antibodies at the indicated dilutions: mouse monoclonal anti-IFT81 (1:400) [29] and anti- ATP synthase β subunit (1:100,000; Agrisera, Vännäs, Sweden, catalog number AS05 085).

### Supporting information

**S1 Appendix. Sequences from *fap70* mutants, donor DNA, and wild type.**
(DOCX)

**S2 Appendix. Alignment of sequences of donor DNA, wild-type DNA, and DNA from *fap70* mutants C6 and D6.**
(DOCX)

**S1 Table. List of crRNA and primer sequences.**
(DOCX)

**S2 Table. Gene identifications.**
(DOCX)

**S1 Fig.**
(PDF)

## Acknowledgments

We thank Dr. Douglas Cole (University of Idaho) for the generous gift of the antibody to *C. reinhardtii* IFT81. We are grateful to Dr. Gregory Pazour (University of Massachusetts Medical School) for advice.

## Author Contributions

**Conceptualization:** Tyler Picariello, Yuqing Hou, George B. Witman.

**Data curation:** Tyler Picariello, Yuqing Hou.

**Formal analysis:** Tyler Picariello, Yuqing Hou, Nathan A. McNeill, Toshiyuki Oda.

**Funding acquisition:** Tomohiro Kubo, Toshiyuki Oda, George B. Witman.

**Investigation:** Tyler Picariello, Yuqing Hou.

**Methodology:** Tyler Picariello, Yuqing Hou, Tomohiro Kubo, Haru-aki Yanagisawa.

**Project administration:** George B. Witman.

**Resources:** Tyler Picariello, Yuqing Hou, George B. Witman.

**Supervision:** George B. Witman.

**Validation:** Tyler Picariello, Yuqing Hou, Tomohiro Kubo, Nathan A. McNeill, Haru-aki Yanagisawa, Toshiyuki Oda.

**Writing – original draft:** Tyler Picariello, Yuqing Hou, George B. Witman.

**Writing – review & editing:** Tyler Picariello, Yuqing Hou, Tomohiro Kubo, Nathan A. McNeill, Haru-aki Yanagisawa, Toshiyuki Oda, George B. Witman.

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
