## [Decision Letter · Decision Letter 0]

6 Feb 2020

PONE-D-20-01228

TIM, a Targeted Insertional Mutagenesis method utilizing CRISPR/Cas9 in Chlamydomonas reinhardtii

PLOS ONE

Dear Prof. Witman,

Thank you for submitting your manuscript to PLOS ONE. After careful consideration, we feel that it has merit but does not fully meet PLOS ONE’s publication criteria as it currently stands. Therefore, we invite you to submit a revised version of the manuscript that addresses the points raised during the review process.

We would appreciate receiving your revised manuscript by Mar 22 2020 11:59PM. To enhance the reproducibility of your results, we recommend that if applicable you deposit your laboratory protocols in protocols.io, where a protocol can be assigned its own identifier (DOI) such that it can be cited independently in the future. For instructions see: http://journals.plos.org/plosone/s/submission-guidelines#loc-laboratory-protocols

We look forward to receiving your revised manuscript.

Kind regards,

Andrew Webber

Academic Editor

PLOS ONE

Additional Editor Comments (if provided):

Thank you for your submission. There are only a few minor revisions suggested by the reviewers. I think these would help enhance an already good manuscript. We look forward to your revisions.

Reviewers' comments:

Reviewer's Responses to Questions

**Comments to the Author**

1. Is the manuscript technically sound, and do the data support the conclusions?

Reviewer #1: Yes

Reviewer #2: Yes

2. Has the statistical analysis been performed appropriately and rigorously? 

Reviewer #1: N/A

Reviewer #2: N/A

3. Have the authors made all data underlying the findings in their manuscript fully available?

Reviewer #1: Yes

Reviewer #2: Yes

4. Is the manuscript presented in an intelligible fashion and written in standard English?

Reviewer #1: Yes

Reviewer #2: Yes

5. Review Comments to the Author

Reviewer #1: This manuscript reports a major advance in Chlamydomonas molecular biology, and will be of great interest to the large and growing community of users who employ Chlamydomonas as a model system. Although it is based on previously published work, cited appropriately, it very carefully works through a large number of important parameters, and in careful fashion establishes the most favorable conditions for gene disruption using Cas9RNPs and CRISPR. The experiments are carefully designed and carefully reported.

The usefulness of the manuscript, however, would be greatly enhanced by an improved presentation of figure 1. First, although the discussion refers to red asterisks and arrows, I do not see these. As the figure is drawn and described this reviewer had a hard time understanding exactly what was done. As drawn, at least to my eyes, the expected PCR product from a perfect insertion would not be centered, or even overlap, the RNP cut site. The "homology arms" added to the antibiotic resistance cassette are not delineated on the figure. Is the left homology arm to the left of the green box? Where are the black arrows representing the homology arms on the top line? Are they in the blue box (target site) or just outside. Perhaps my confusion comes from the labeling of the green boxes on the second and third lines. The second box refers to the antibiotic resistance cassette with homology arms, and the third line just refers to the antibiotic resistance cassette. The boxes, however, are the same size. On the NHEJ side of the figure, what is the orange box? Is it formed from the remaining parts of the target?

It would be very helpful for understanding the nature of the insertions to have some of the PCR products sequenced and compared to the starting genome. Were the cuts precise, or ragged? Some of the ca. 2 kb bands appear to be doublets, or somewhat offset from the others. Alternatively a second round of PCR using a primer within the antibiotic resistance gene and one of the outside primers would be useful for characterizing the inserts. Of particular interest with regard to the question of the structure of insertion sites is the experiment on line 18 of Table 1. Even without added homology arms, the insertion cassette was present at the cut site in 20 of 48 drug-resistant transformants. These insertions have to be a result of NHEJ, because there is no homology to target the insertion event. Thus in almost half of events cells chose to "grab" the ends of the cassette to fill that gap rather than just grabbing the broken ends of the chromosome. If this observation held with more experiments on other genes, a user of this technique might not choose to "customize" their cassette with homology arms if targeting was achieved just choosing by the specificity of the Cas9 cut site.

A smaller improvement in the manuscript that should be included is to drop the mention of the experiment on rows 12 and 13 of Table 1. Because autolysin treatment of these cells was ineffective, as noted in the legend, there is no information to be gained about whether 24 vs. 48 hours of growth before selection is preferred.

Reviewer #2: This manuscript describes methods for targeted insertional mutagenesis (TIM) in the unicellular green alga, Chlamydomonas reinhardtii. The organism has proved to be an important model system for studies of multiple processes and organelles, including cilia and flagella, the cell cycle, fertilization, photosynthesis, and phototaxis. As the authors indicate, an excellent mutant library is now available for the community, but a reliable, efficient method for targeted mutagenesis has been needed for study of many genes not available in the library. The manuscript provides compelling evidence that by use of these optimized methods, investigators can now generate desired mutants at high efficiency. The manuscript is especially well-written and was a pleasure to read.

The only minor suggestion I have is that the authors clarify the steps in their procedure in which cells are in the M-N medium. LIne 317 states “During experiments, cells were placed in M-N medium (modified M medium lacking NH4NO3).” Apparently, the autolysin is in M-N medium, and during autolysin treatment, cells will be in M-N. But, according to the text, before autolysin treatment the cells are in growth medium, and after autolysin, the cells are washed from that solution into other media.

6. PLOS authors have the option to publish the peer review history of their article (what does this mean?). If published, this will include your full peer review and any attached files.

Reviewer #1: No

Reviewer #2: No

---

## [Author Response · Author response to Decision Letter 0]

6 Mar 2020

Reviewer #1:

The usefulness of the manuscript, however, would be greatly enhanced by an improved presentation of figure 1. First, although the discussion refers to red asterisks and arrows, I do not see these. 

Response: To make it easier to find the asterisks and arrows, we now specify in the legend that they are “on the right side of the gel.” We also have changed their color to black and have made them slightly larger.

As the figure is drawn and described this reviewer had a hard time understanding exactly what was done. As drawn, at least to my eyes, the expected PCR product from a perfect insertion would not be centered, or even overlap, the RNP cut site. The "homology arms" added to the antibiotic resistance cassette are not delineated on the figure. Is the left homology arm to the left of the green box? Where are the black arrows representing the homology arms on the top line? Are they in the blue box (target site) or just outside. Perhaps my confusion comes from the labeling of the green boxes on the second and third lines. The second box refers to the antibiotic resistance cassette with homology arms, and the third line just refers to the antibiotic resistance cassette. The boxes, however, are the same size. 

Response: We are grateful for the feedback on the shortcomings of the original figure. The schematic diagram of Figure 1A has now been modified in several ways. Importantly, we have now drawn the diagrams in the first and second rows to the same scale as those in the third row. We now show the positions of the homology arms on the donor DNA, and of the corresponding sequences on the genome; these are marked with L for left and R for right. A branch has been added to the arrow in the (left) pathway to indicate how the antibiotic-resistance cassette is inserted into the genome in that pathway. We have added black arrows to indicate the positions of the primers used for screening. 

On the NHEJ side of the figure, what is the orange box? Is it formed from the remaining parts of the target?

Response: In the right pathway, we have changed “NHEJ causing indel mutation” to “End joining without integration of donor DNA,” which is more inclusive of the events that might occur in this pathway. The orange box was indeed meant to indicate the remaining parts of the target; we have now replaced it with a smaller box, colored blue to match the target site from which it is derived. A small white box is inserted into the small blue box to indicate that if an indel occurs in this pathway, it is small and will not change the size of the PCR product. Arrows with labels have been added to show the positions of the primers relative to the target site.

The figure legend has been revised to reflect these changes, which have improved the figure immensely. 

It would be very helpful for understanding the nature of the insertions to have some of the PCR products sequenced and compared to the starting genome. Were the cuts precise, or ragged? Some of the ca. 2 kb bands appear to be doublets, or somewhat offset from the others. Alternatively a second round of PCR using a primer within the antibiotic resistance gene and one of the outside primers would be useful for characterizing the inserts. 

Response: We appreciate the reviewer bringing this to our attention. We also were curious about the nature of the mutations, so we randomly picked and sequenced seven larger PCR bands from fap70 mutants generated using 50-bp homology arms. In wild type, sequence corresponding to the left homology arm of the donor DNA is 27 bps upstream of the PAM, while sequence corresponding to the right homology arm is 3 bps downstream from the PAM. Five of the mutants had sequences identical to that of wild type except that the 33 bps between the sequences corresponding to the homology arms were replaced with an intact paromomycin-resistance cassette, as expected for HDR. In each of the other two mutants, partial cassette sequence with 5’ and 3’ indels was similarly present between regions corresponding to the left and right homology arms. We have added a paragraph to the Results section describing the sequencing data and briefly discuss the new data in the Discussion section. We have added the sequencing method to the Methods and Materials, and have added one primer used in sequencing to Table S1. We also have added two supplementary files (Appendix S1 reporting the mutant sequences and Appendix S2 showing the alignment of the sequences) for readers who might be interested in the details. The data do not enable conclusions regarding whether the cuts were precise or ragged. However, the sequencing did confirm the insertion of our donor DNA at the intended target site.

Of particular interest with regard to the question of the structure of insertion sites is the experiment on line 18 of Table 1. Even without added homology arms, the insertion cassette was present at the cut site in 20 of 48 drug-resistant transformants. These insertions have to be a result of NHEJ, because there is no homology to target the insertion event. Thus in almost half of events cells chose to "grab" the ends of the cassette to fill that gap rather than just grabbing the broken ends of the chromosome. If this observation held with more experiments on other genes, a user of this technique might not choose to "customize" their cassette with homology arms if targeting was achieved just choosing by the specificity of the Cas9 cut site.

Response: We agree with the reviewer that these insertions are most likely a result of NHEJ. However, with all due respect, the fact that 20 out of 48 drug-resistant transformants had insertions at the target site does not mean that “in almost half of events cells chose to ‘grab’ the ends of the cassette to fill that gap rather than just grabbing the broken ends of the chromosome.” The other 28 antibiotic-resistant strains that did not pass our PCR screen presumably represent a subset of cells in which the antibiotic-resistance cassette integrated elsewhere in the genome. These cells may or may not have been cut at the target site by the RNP. Additionally, we do not know the fraction of cells that were cut at the target site and did not undergo insertion of donor DNA. This fraction could have been large, in which case only a small (but unknown) fraction of the cells with double-strand breaks at the target site underwent integration of the donor DNA lacking homology arms. Therefore, it is not possible to determine from the data what fraction of cells with double-strand breaks chose to grab the ends of the cassette to fill the gap.

Nevertheless, the fact that we were able to select for many correctly targeted insertional mutants even in the absence of homology arms is important. We have revised the relevant paragraph of the Discussion to highlight this point and the evidence that inclusion of homology arms is advantageous. We thank the reviewer for calling this to our attention.

A smaller improvement in the manuscript that should be included is to drop the mention of the experiment on rows 12 and 13 of Table 1. Because autolysin treatment of these cells was ineffective, as noted in the legend, there is no information to be gained about whether 24 vs. 48 hours of growth before selection is preferred.

Response: We thank the reviewer for this suggestion. Rows 12 and 13 of original Table 1 have been deleted, and the subsequent rows renumbered accordingly in both the table and text. The paragraph discussing the results of the experiment in original rows 12 and 13 has been deleted.

Reviewer #2:

The only minor suggestion I have is that the authors clarify the steps in their procedure in which cells are in the M-N medium. LIne 317 states “During experiments, cells were placed in M-N medium (modified M medium lacking NH4NO3).” Apparently, the autolysin is in M-N medium, and during autolysin treatment, cells will be in M-N. But, according to the text, before autolysin treatment the cells are in growth medium, and after autolysin, the cells are washed from that solution into other media.

Response: The reviewer’s interpretation is correct, and we thank him or her for pointing out that we introduced confusion with the wording in the original Materials and Methods. We have now changed the quoted sentence to simply “Media used are: Tris-acetate-phosphate (TAP) medium [41]; M (minimal) medium I [42] modified to contain 0.0022 M KH2PO4 and 0.00171 M K2HPO4 [43]; M-N medium (modified M medium lacking NH4NO3).” Under “Autolysin preparation,” we state that cells were scrapped from plates and resuspended in M-N medium until gametes formed. As the reviewer notes, autolysin was in M-N medium (this should be clear from the description of “Autolysin preparation”), and cells were briefly treated with this solution to remove their cell walls (described under “Delivery of Cas9/gRNA RNP and donor DNA”).

Finally, we have moved the final paragraph of the Results section to the end of the Discussion, which is a more appropriate place for it, improved what was the final paragraph but is now the penultimate paragraph of the Discussion, and made a few very minor changes elsewhere, e.g., to report which exons were targeted and to improve syntax or clarity.

---

## [Decision Letter · Decision Letter 1]

20 Apr 2020

TIM, a Targeted Insertional Mutagenesis method utilizing CRISPR/Cas9 in Chlamydomonas reinhardtii

PONE-D-20-01228R1

Dear Dr. Witman,

We are pleased to inform you that your manuscript has been judged scientifically suitable for publication and will be formally accepted for publication once it complies with all outstanding technical requirements.

With kind regards,

Hodaka Fujii, M.D., Ph.D.

Academic Editor

PLOS ONE

Additional Editor Comments (optional):

Reviewers' comments:

Reviewer's Responses to Questions

**Comments to the Author**

1. If the authors have adequately addressed your comments raised in a previous round of review and you feel that this manuscript is now acceptable for publication, you may indicate that here to bypass the “Comments to the Author” section, enter your conflict of interest statement in the “Confidential to Editor” section, and submit your "Accept" recommendation.

Reviewer #1: All comments have been addressed

2. Is the manuscript technically sound, and do the data support the conclusions?

Reviewer #1: Yes

3. Has the statistical analysis been performed appropriately and rigorously? 

Reviewer #1: Yes

4. Have the authors made all data underlying the findings in their manuscript fully available?

Reviewer #1: Yes

5. Is the manuscript presented in an intelligible fashion and written in standard English?

Reviewer #1: Yes

6. Review Comments to the Author

Reviewer #1: (No Response)

7. PLOS authors have the option to publish the peer review history of their article (what does this mean?). If published, this will include your full peer review and any attached files.

Reviewer #1: No

---

## [Editor Report · Acceptance letter]

28 Apr 2020

PONE-D-20-01228R1 

TIM, a Targeted Insertional Mutagenesis method utilizing CRISPR/Cas9 in *Chlamydomonas reinhardtii*

Dear Dr. Witman:

I am pleased to inform you that your manuscript has been deemed suitable for publication in PLOS ONE. Congratulations! Your manuscript is now with our production department. 

With kind regards,

on behalf of

Dr. Hodaka Fujii 

Academic Editor

PLOS ONE